# Multifaceted interventions for healthcare-associated infections and rational use of antibiotics in a low-to-middle-income country: Can they be sustained?

Indah K. Murni[1,2]*, Trevor Duke[3,4], Sharon Kinney[5], Andrew J. Daley[4,6], Ida S. Laksanawati[1,2], Nurnaningsih[1], Desy Rusmawatiningtyas[1], M. Taufik Wirawan[1], Yati Soenarto[1,2]

1 Department of Pediatrics, DR. Sardjito Hospital/Faculty of Medicine, Public Health and Nursing, Universitas Gadjah Mada, Yogyakarta, Indonesia, 2 Pediatric Research Office, Faculty of Medicine, Public Health and Nursing, Universitas Gadjah Mada, Yogyakarta, Indonesia, 3 Centre for International Child Health, Department of Pediatrics, Royal Children's Hospital, University of Melbourne, MCRI, and Pediatric Intensive Care Unit, Melbourne, Australia, 4 Department of Pediatrics, University of Melbourne, Melbourne, Australia, 5 Department of Pediatrics and Nursing, Royal Children's Hospital, University of Melbourne, Melbourne, Australia, 6 Infection Prevention and Control, Royal Children's Hospital, Melbourne, Australia

* indah.kartika.m@ugm.ac.id

**Data Availability Statement:** All relevant data are within the manuscript and its Supporting Information files.

## Abstract

### Background

Transmission of infection between patients by health workers, and the irrational use of antibiotics are preventable causes for healthcare-associated infections (HAI) and multi-resistant organisms. A previous study implementing a hand hygiene campaign and antibiotic stewardship program significantly reduced these infections. Sustaining such interventions can be challenging.

### Aims

To evaluate whether there was a sustained effect of a multifaceted infection control and antibiotic stewardship program on HAI and antibiotic use 5 years after it began.

### Methods

A prospective evaluation study was conducted over 26 months (from February 2016 to April 2018) in a teaching hospital in Indonesia, 5 years after the implementation of an antibiotic stewardship and infection control program, which was successful when initially evaluated. All children admitted to the pediatric ICU and pediatric wards were observed daily. Assessment of HAI was made based on the criteria from the Centers for Disease Control and Prevention. Assessment of rational antibiotic use was based on the WHO Pocket Book of Hospital Care for Children. Multivariable logistic regression analysis was used to quantify the relationship between the HAI and the multifaceted intervention.

**Funding:** The authors received no specific funding for this work.

**Competing interests:** The authors have declared that no competing interests exist.

## Results

We observed an increase in HAIs, from 8.6% (123/1419 patients) in the initial post-intervention period in 2011–2013 to 16.9% (314/1855) in the evaluation study (relative risk (RR) (95% CI) 1.95 (1.60 to 2.37)). After adjusting for potential confounders, we found that an increase in HAI in the evaluation period with adjusted OR 1.94 (95% CI 1.53 to 2.45). Inappropriate antibiotic use also increased, from 20.6% (182 of 882 patients who were prescribed antibiotics) to 48.6% (545/1855) (RR 2.35 (2.04 to 2.71)). Hand hygiene compliance also declined from 62.9% (1125/1789) observed moments requiring hand hygiene to 51% (1526/2993) (RR 3.33 (2.99 to 3.70)).

## Conclusions

Healthcare-associated infections and irrational use of antibiotics remain significant even after the implementation of a multifaceted infection control intervention and antibiotic stewardship program. There is a need for continuous input, ongoing surveillance and long-term monitoring of these interventions to sustain compliance and effectiveness and address problems as they emerge.

## Introduction

Morbidity and mortality of healthcare-associated infections in children in low-to-middle-income countries are significant, and these can be reduced with a program of low cost interventions [1,2]. The high burden of healthcare-associated infections and the significant effect of the interventions on both healthcare-associated infections and in-hospital mortality requires effective policy and practice responses to prevent these infections [3]. Healthcare-associated infections should never be considered as inevitable, even in settings with limited resources.

In Yogyakarta, Indonesia, we conducted surveillance for healthcare-associated infection and long-term monitoring of the effectiveness of the multifaceted intervention described in the previous published study [1]. This current study provides an evaluation of the incidence of healthcare-associated infection, irrational antibiotic use, hand hygiene compliance, and mortality to evaluate whether the effect of the multifaceted infection control intervention was sustained over 5 years.

## Material and methods

### Setting

The study was conducted at the Dr. Sardjito Teaching Hospital, Yogyakarta, Indonesia, in the Pediatric Intensive Care Unit (PICU) and the public general pediatric wards. The Dr. Sardjito Hospital is a referral hospital for Yogyakarta and the Southern part of the Central Java province in Indonesia, and provides services to a population of approximately 3.6 million people.

### Design

A prospective cohort study evaluating whether the previous multifaceted intervention to reduce healthcare-associated infections was sustained after 5 years [1].

### Inclusion criteria

Patients who were expected to remain in the pediatric wards or PICU for more than 48 hours.

### Outcome measures and data collection

Data collection methods for this study were similar to a previous effectiveness study on reducing healthcare-associated infections and improving rational use of antibiotics [1].

**A. Healthcare-associated infection.** The definitions of healthcare-associated infection were based on the US Centers for Disease Control and Prevention (CDC) National Healthcare Safety Network (NHSN) [4,5]. Every child in the study was observed each day to see whether he/she fulfilled the CDC criteria for a healthcare-associated infection. If criteria were fulfilled, the treating medical and nursing staff were advised to collect a culture of blood, urine or other sterile sites as appropriate on the same day.

We did not have data detailing which cultures were collected based on the study surveillance definition or based on clinical suspicion of infection of the treating doctors. We included those two as one. But we did an identical data collection to diagnose nosocomial infections in both periods. Those cultures were taken when patients had signs and symptoms of infection.

**B. The rational use of antibiotics.** Whether antibiotic use was rational or inappropriate was assessed at the time of study entry in every patient with a community-acquired infection who was treated with antibiotics, and each day during their hospital admission. The standards for empirical antibiotic prescribing were based on the recommendations contained in the WHO *Pocket Book of Hospital Care for Children* [6]. Patients had antibiotic use recorded daily from their medical record. Inappropriate antibiotic use was classified according to the spectrum, dose, and duration. Inappropriate spectrum was defined if a child received antibiotics inconsistent with the standard guideline, or broader spectrum antibiotics than the recommendation, or was exposed to unnecessary therapeutic or prophylactic antibiotics. Inappropriate dose was defined if a child received antibiotics at 20% more or less than the WHO recommended dose, or if there was insufficient dosage adjustment in renal or hepatic insufficiency [7]. Inappropriate duration was defined as antibiotic used for more than 20% longer than the recommended duration in the standard without a documented reason.

**C. Hand hygiene compliance.** Hand hygiene compliance was defined as hand washing with antiseptic soap and water or alcohol-based hand rubs for each of the World Health Organization's five moments for hand hygiene [8,9]. Hand hygiene compliance was achieved when there was an indication for hand hygiene and the health worker performed this correctly. Health workers (doctors and nurses) were randomly chosen for observation using a fixed time of observation (20 ± 10 minutes each). During these periods of observations the first health worker who was involved in the care of the patient was selected [10]. Direct hand hygiene observation began when the health worker entered the patient's room or bed area and was observed during activities that involved contact with the patient or their environment and observation ended when the health worker completed the activity and left the bed space. Health workers were informed at the beginning of the project about the hand hygiene audits, but they were not told they were specifically being observed. However, they were familiar with the role of the researcher and research assistant, whose presence was not hidden within the ward environment.

*Validation of bacterial culture as a measure of healthcare-associated infection*. We used standard culture techniques that had been validated in our hospital using BACTEC® 9120 (BD Diagnostics, Sparks, MD, USA). Bacterial isolation and antibiotic susceptibility testing were performed according to Clinical Pathology standard procedures [11]. For each positive result, the type of isolated organism, number of positive culture sites, time to culture positivity, the

presence of focal or generalized clinical signs of infection and an overall assessment of illness were recorded. This enabled an assessment of whether the isolate was a true pathogen or a contaminant [12].

## Multifaceted intervention

During 2013 to 2018, the infection control and antibiotic stewardship program had been continually implemented using the same model as in the previous published study [1]. The hospital had also achieved accreditation from the Joint Commission International (https://www.jointcommissioninternational.org). The requirement of this accreditation includes developing and implementing a program for infection control and antibiotic stewardship.

The infection control and antibiotic stewardship team consist of infectious disease doctors, specialist doctors, nurses, the pharmacist and clinical pathology and microbiology staff. We have no protected time for infection control and antibiotic stewardship nor do health care professionals involved receive additional salary for these roles.

Activities to contain healthcare-associated infections and irrational antibiotic use in Dr Sardjito Hospital included: (1) Establishing effective hospital therapeutics committees with the responsibility for overseeing antibiotic use; (2) developing guidelines for antibiotic treatment and prophylaxis and antibiotic formularies; (3) monitoring antibiotic use and feedback to prescribers; (4) establishing infection control programs for effective management of antibiotic resistance; (5) ensuring performance and quality assurance of pathogen identification and antibiotic susceptibility tests; and (6) controlling and monitoring pharmaceutical company promotional activities.

In the post-intervention and evaluation phase the team collected surveillance data on healthcare-associated infection and antibiotic use prospectively, but the surveillance did not become fully incorporated into routine quality activities. We did not have team conducting active surveillance; they collected data by passive surveillance of identified healthcare-associated infection. The hospital did not perform surveillance of antibiotic use and rational use of antibiotics. While some educational activities were conducted, audit-feedback activities were not.

Some monitoring of antibiotic use was continued routinely in the follow-up phase. For example, we discussed pediatric patients with antibiotic-related problems every week in the wards and PICU. We monitored and reported on pathogens identified and antibiotic susceptibility every 3 to 6 months. We also restricted certain antibiotics such as carbapenems, so that these antibiotics needed approval by an Antibiotic Stewardship Committee. However, monitoring antibiotic use and feedback to prescribers were not done routinely. The detailed interventions in the intervention, post-intervention, and evaluation periods were described in Table 1.

## Outcome measures

The primary outcome was the incidence of patients with healthcare-associated infection. The incidence rate of healthcare-associated infection is reported according to two standard metrics: number of patients who developed a healthcare-associated infection/100 patients (the proportion of patients who developed a healthcare-associated infection) and healthcare-associated infection episodes/1000 patient-days.

Secondary outcomes were the proportions of patients who were exposed to the irrational use of antibiotics; hand hygiene compliance among healthcare workers; and all-cause in mortality rates.

**Table 1. Interventions and activities done in each period of the study.**

| | Intervention period (December 2010 to February 2011) | Post-intervention period (March 2011 –February 2013) and Evaluation period (February 2016 –February 2018) |
|---|---|---|
| Type of intervention | Educational seminars, reminders (handy module, CD, antibiotic chart), audit, and performance feedback | Ongoing education was provided where needed |
| Timing of interventions | Seminars were conducted at least twice for each topic for one hour, to cover all the health workers on different shifts. | Seminars were conducted only when there were new staff, pediatric residents, or medical clerkship students. |
| | The surveillance or audit data were collected prospectively and fed back to the health workers individually and were presented at the monthly meetings. | The surveillance or audit data were collected prospectively, but not fed back to the health workers. |
| Hand hygiene | Hand hygiene campaign was done routinely. | Hand hygiene campaign was done routinely. |
| | | The hospital had achieved accreditation from the Joint Commission International |
| Resources for hand hygiene | A bottle of alcohol hand rub using WHO recommended formula had already been made available in every patient care room and another bottle was placed at the entrance of each room. | A bottle of alcohol hand rub using WHO recommended formula had already been made available in every patient care room and another bottle was placed at the entrance of each room. |
| | There was a water sink and antiseptic soap in every ward. | There was a water sink and antiseptic soap in every ward. |
| Antibiotic stewardship program | Activities to contain nosocomial infections and irrational antibiotic use included: (1) Establishing effective hospital therapeutics committees with the responsibility for overseeing antibiotic use; (2) developing guidelines for antibiotic treatment and prophylaxis and antibiotic formularies; (3) monitoring antibiotic use and feedback to prescribers; (4) establishing infection control programs for effective management of antibiotic resistance; (5) ensuring performance and quality assurance of pathogen identification and antibiotic susceptibility tests; and (6) controlling and monitoring pharmaceutical company promotional activities. | Activities to contain nosocomial infections and irrational antibiotic use included: (1) Establishing effective hospital therapeutics committees with the responsibility for overseeing antibiotic use; (2) developing guidelines for antibiotic treatment and prophylaxis and antibiotic formularies; (3) monitoring antibiotic use and feedback to prescribers; (4) establishing infection control programs for effective management of antibiotic resistance; (5) ensuring performance and quality assurance of pathogen identification and antibiotic susceptibility tests; and (6) controlling and monitoring pharmaceutical company promotional activities. |
| Timing of antibiotic stewardship program | Discussion on pediatric patients with antibiotic-related problems every week in the wards or PICU, antibiotic restriction, and monitoring antibiotic use and feedback to prescribers were done routinely. | Discussion on pediatric patients with antibiotic-related problems every week in the wards or PICU was routinely done. Antibiotic restriction was routinely implemented. |
| | Monitoring pathogen identification and antibiotic susceptibility tests were reported every 3 to 6 months. | Monitoring pathogen identification and antibiotic susceptibility tests were reported every 3 to 6 months. |
| | | Monitoring antibiotic use was done prospectively, but feedback to prescribers were not done routinely. |
| Other measures | Measures to prevent nosocomial bloodstream infections, ventilator-associated pneumonia, and catheter-associated urinary tract infections were implemented. | Measures to prevent nosocomial bloodstream infections, ventilator-associated pneumonia, and catheter-associated urinary tract infections were implemented. |
| Composition/staffing of the infection control and antimicrobial stewardship programs | Infectious disease doctors, specialist doctors, nurses, the pharmacist and clinical pathology and microbiology staff. No protected time for infection control and antibiotic stewardship nor do health care professionals involved receive additional salary for these roles. | Infectious disease doctors, specialist doctors, nurses, the pharmacist and clinical pathology and microbiology staff. No protected time for infection control and antibiotic stewardship nor do health care professionals involved receive additional salary for these roles. |

## Data analysis

After transfer into Excel, data were analyzed using STATA version 12.1 (StataCorp LP, Texas). The $\chi^2$ statistic was used to analyze the results when comparing proportions from both periods. A probability value $< 0.05$ was considered to denote statistical significance. The relative risk (RR) was calculated to compare the effect of the interventions between both periods by calculating the ratio of the probability of an outcome in the post-intervention period to the probability of an outcome in the evaluation period. Multivariable logistic regression analysis was used to quantify the relationship between the healthcare-associated infection and the multifaceted intervention allowing for statistical control of potential confounders including age, sex, the presence of syndrome, immunocompromised condition, referral patient, sepsis, and

malnutrition. These variables were added stepwise and included in the model when having a probability value of < 0.10. The Ethics Committees of the Universitas Gadjah Mada approved the study. The ethics committee did not require individual patient consent, but all parents of children were informed of the study.

## Results

A total of 1855 patients were recruited during the follow-up evaluation period from February 2016 to April 2018. Patients in the post-intervention and the follow-up evaluation periods were similar with regard to age, proportion admitted to ICU, and some underlying diseases, although there was a higher proportion of patients referred from another hospital during the follow-up period (Table 2).

### The incidence of healthcare-associated infection

The risk of a patient developing a healthcare-associated infection increased from 8.6% (95% confidence intervals [CI] 7.3–10.2%) in the post-intervention period to 16.9% in the follow-up

**Table 2. Baseline characteristics of patients in the post-intervention and the evaluation period.**

| Characteristics | Post-intervention n = 1419 (%) | Evaluation period n = 1855 (%) |
|---|---|---|
| Male sex–n (%) | 797 (56.1) | 943 (50.8) |
| Age–n (%) | | |
| ≤ 12 months | 351 (24.7) | 483 (26.1) |
| > 12–60 months | 365 (25.7) | 479 (25.8) |
| > 60–120 months | 327 (23.0) | 366 (19.7) |
| > 120 months | 376 (26.5) | 527 (28.4) |
| Source of patients–n (%) | | |
| Community | 835 (58.8) | 774 (41.7) |
| Referral patients | 492 (34.6) | 975 (52.6) |
| Transferred from other units within hospital | 92 (6.4) | 106 (5.7) |
| Ward or setting–n (%) | | |
| PICU | 281 (19.8) | 286 (15.4) |
| General pediatric wards | | |
| Infectious ward | 450 (31.7) | 538 (29) |
| Non-infectious ward | 688 (48.4) | 1031 (55.6) |
| Underlying diseases–n (%) | | |
| Neurology | 229 (16.1) | 338 (18.2) |
| Nephrology | 121 (8.5) | 179 (9.7) |
| Respiratory | 169 (11.9) | 153 (8.3) |
| Cardiovascular | 187 (13.1) | 478 (25.8) |
| Hematology and oncology | 177 (12.4) | 17 (0.9) |
| Gastrohepatology | 147 (10.3) | 236 (12.7) |
| Infectious | 89 (6.2) | 209 (11.3) |
| Immunology | 107 (7.5) | 123 (6.6) |
| Sepsis | 71 (5) | 49 (2.6) |
| Endocrinology | 22 (1.5) | 26 (1.4) |
| Malnutrition | 12 (0.8) | 7 (0.4) |
| Surgery | 88 (6.2) | 40 (2.2) |

The post-intervention period was from March 2011 to February 2013. The evaluation period was from February 2016 to April 2018.

evaluation period: RR 1.95 (95% CI 1.60–2.37) (Table 3). There was an increase in the incidence rate of healthcare-associated infection from 9.3 per 1000 patient days (125/13498) to 20.1 infections per 1000 patient-days (416/20672). Healthcare-associated infection rates had increased 5 years after the initiation of what initially was a successful multifaceted intervention (Fig 1).

After adjusting for potential confounders, we found an increase in healthcare-associated infections in the evaluation period with adjusted OR 1.94 (95% CI 1.53 to 2.45).

The incidence of healthcare-associated bloodstream infection was 92/1419 (6.5%) among all recruited children and 92/123 (74.8%) among children with healthcare-associated infection in the post intervention period, while in the evaluation period was 36/1885 (1.9%) among all recruited children or 36/314 (11.5%) among children with healthcare-associated infection. In the post-intervention period, the most common organism related to healthcare-associated infection was *Pseudomonas aeruginosa*, while in the evaluation period was *Klebsiella pneumoniae.*

### The irrational use of antibiotics

The overall use of antibiotics was not different in the post-intervention and follow-up evaluation periods; these were prescribed for 62.2% (882/1419) and 60.5% (1122/1855) of all patients respectively (p = 0.33). However, compared with the post-intervention period, in the follow-up evaluation period the risk of patients being exposed to irrational or inappropriate antibiotics increased from 20.6% (182/882) to 48.6% (545/1855), respectively, with RR 2.35 (95% CI 2.04–2.71) (Table 4).

### The compliance of hand hygiene among health workers

Hand hygiene compliance decreased significantly in the follow-up evaluation period in PICU and the general non-infectious wards (Table 5). Overall hand hygiene compliance among the healthcare workers in the evaluation period decreased from 62.9% (1125/1789) to 51% (1526/2993) (p<0.001).

### Mortality

The risk of in-hospital mortality among children in the hospital decreased by 26% in the follow-up evaluation period compared to the post-intervention period, from 8.0% (114/1419) to 6.9% (113/1855) with RR 0.74 (95% CI 0.57–0.97). Mortality in children with healthcare-associated infection was 4.1% (13/314) in the evaluation period. Mortality associated with

**Table 3. The incidence of healthcare-associated infection over the study period.**

|  | Incidence of HAI | | Relative risk (95%CI) |
|---|---|---|---|
|  | Post-intervention (%) | Evaluation (%) |  |
| Pediatric ICU | 48/281 (17) | 76/286 (26.6) | 1.55 (1.13–2.14) |
| General infectious ward | 44/450 (9.7) | 100/530 (18.7) | 1.93 (1.38–2.68) |
| General non-infectious ward | 31/688 (4.5) | 138/1038 (13.4) | 2.95 (2.02–4.30) |
| Overall | 123/1419 (8.6) | 314/1855 (16.9) | 1.95 (1.60–2.37) |

HAI = healthcare-associated infection

The relative risk was derived by calculating the ratio of the probability of an outcome in the post-intervention period to the probability of an outcome in the evaluation period.

The post-intervention period was from March 2011 to February 2013. The evaluation period was from February 2016 to April 2018

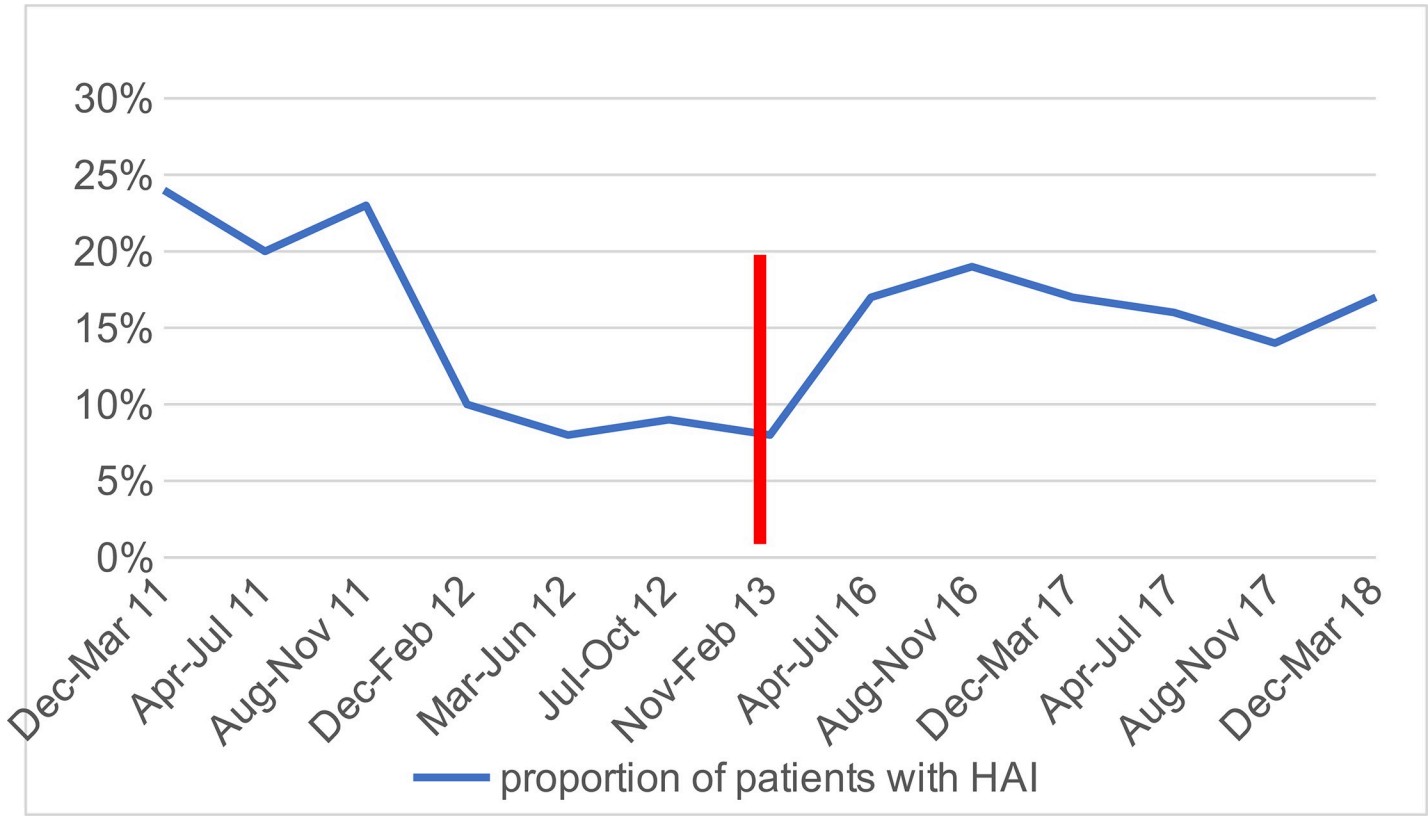

**Fig 1. The proportion of healthcare-associated infections in the intervention and post-intervention era (December 2011 to February 2013) and the follow up evaluation era (started in February 2016), the gap between those periods is indicated with the red line.**

**Table 4. The incidence of antibiotic use and irrational use of antibiotics over the study period.**

| | Pediatric ICU | | Relative risk (95% CI) | Infectious ward | | Relative risk (95% CI) | Non-infectious ward | | Relative risk (95% CI) | Overall | | Relative risk (95% CI) |
|---|---|---|---|---|---|---|---|---|---|---|---|---|
| | Post n = 281 (%) | Evaluation n = 286(%) | | Post n = 450 (%) | Evaluation n = 538 (%) | | Post n = 688 (%) | Evaluation n = 1031(%) | | Post n = 1419 (%) | Evaluation n = 1855(%) | |
| Number of patients with antibiotics | 258 | 219 | | 340 | 365 | | 284 | 538 | | 882 | 1122 | |
| Number of patients exposed to incorrect or inappropriate antibiotics | 93 (36) | 172 (78.5) | 2.18 (1.82–2.59) | 52 (15.3) | 192 (52.6) | 3.44 (2.63–4.49) | 37 (13) | 181 (33.6) | 2.58 (1.87–3.56) | 182 (20.6) | 545 (48.6) | 2.35 (2.04–2.71) |
| - Inappropriate spectrum | 89 | 162 | | 52 | 173 | | 34 | 175 | | 175 | 510 | |
| - Incorrect dose | 0 | 10 | | 0 | 19 | | 2 | 6 | | 2 | 35 | |
| - Inappropriate duration | 1 | 0 | | 0 | 0 | | 0 | 0 | | 1 | 0 | |

The relative risk was derived by calculating the ratio of the probability of an outcome in the post-intervention period to the probability of an outcome in the evaluation period.

The post-intervention period was from March 2011 to February 2013. The evaluation period was from February 2016 to April

**Table 5. The hand hygiene compliance among healthcare workers over the study period.**

| | Compliance with hand hygiene | | p value | Relative risk (95%CI) |
|---|---|---|---|---|
| | Post-intervention (%) | Evaluation (%) | | |
| Pediatric ICU | 390/625 (62.4) | 271/714 (37.9) | < 0.001 | 0.61 (0.54–0.68) |
| General infectious ward | 356/598 (59.5) | 575/1050 (54.8) | 0.056 | 0.92 (0.84–1.00) |
| General non-infectious ward | 379/566 (66.9) | 680/1229 (55.3) | < 0.001 | 0.83 (0.77–0.90) |
| Overall | 1125/1789 (62.9) | 1526/2993 (51) | <0.001 | 0.81 (0.77–0.85) |

The relative risk was derived by calculating the ratio of the probability of an outcome in the post-intervention period to the probability of an outcome in the evaluation period.

The post-intervention period was from March 2011 to February 2013. The evaluation period was from February 2016 to April 2018.

healthcare-associated infection was markedly lower in this evaluation period compared to the mortality in the post-intervention period of 24.5%.

## Discussion

### Major findings

Surveillance of healthcare-associated infection and the irrational use of antibiotics was conducted to determine whether the program instituted in 2010–2013 had a sustained effect on rates of healthcare-associated infection and the irrational use of antibiotics in our large teaching hospital [1]. Although the infection control program had become an integral part of the health culture in our hospital, this current study showed that the clinical impact of the multifaceted infection prevention strategy was not fully sustained. There were no major differences in the patient population in the two time periods, and even after adjusting for differences, there was increase in healthcare-associated infections.

The trend of healthcare-associated infection rates had increased 5 years after the initiation of the multifaceted intervention, despite the program being highly successful in the first 18 months of detailed observation. The cumulative incidence of healthcare-associated infection in the current era study population was 16.9% of patients surveyed, with an incidence rate of 20.1 infections per 1000 patient-days; higher than in the same setting after initiating a multifaceted intervention on infection control and antibiotic stewardship program 5 years before. The earlier multifaceted intervention had brought the incidence down to 8.7% or 9.3 infections per 1000 patient-days [1]. Staff turnover and health care shortages create problems because new doctors and nurses were not aware of infection control bundle especially in PICU. Increasing nurse-to-patient ratios have been associated with the spread of infection. Further, nurse shortages are also related to increased risk of healthcare-associated infections [13].

The irrational use of antibiotic was high compared to the previous published study in the same setting. Some interventions were done routinely such as restricting antibiotic availability by our Antibiotic Stewardship Committee, so that they had to approve use of expensive broad-spectrum agents. Weekly review of antibiotic use for 1–2 patients in wards and PICU was also done. However, antibiotic stewardship rounds as an active process for all patients on antibiotics were not performed routinely. Further, monitoring of antibiotic use and feedback to prescribers were also not done routinely. In our pediatric wards and PICU we indeed had antibiotic guidelines for many years. But our patients mostly came from other hospitals and those patients had been given "high class of antibiotics" and most doctors at our hospital considered that the antibiotics needed to be escalated. Therefore, we need to start a weekly antibiotic stewardship ward round promoting more disciplined and rational antibiotic prescribing,

weekly reminders of best practice in prescribing, enforcing the rules, ceasing or scaling back antibiotics, putting in stop dates in the prescription, and enforcing the writing of indications. These activities are only effective if done routinely on a weekly basis [14].

This finding also highlights the partially sustained effect of intervention of hand hygiene campaign in our hospital. There is a need for ongoing interventions to improve the compliance of hand hygiene practice among the healthcare workers. Although the hand hygiene campaign was in place as a part of the hospital program, education for healthcare workers on the rationale for and how to conduct hand hygiene practice can reduce healthcare-associated infection, and is routinely needed to reduce potentially significant burden of infection especially in a setting like our hospital when the turnover of health workers exists.

Both overall in-hospital mortality in children in the study during this follow-up evaluation period and mortality related to healthcare-associated infection were lower in this evaluation period, compared to the previous published study [1]. One of the possible reasons for the lower mortality rates could be due to a secular trend to lower death rates in hospitals that might relate to improved overall quality of care, preventative strategies, better overall nutrition, and substantial changes in epidemiology including fewer bacterial sepsis and more viral infections [15].

## Comparisons with other studies and explanation for findings

There are limited published studies on the sustainability of the effectiveness of interventions on infection control and antibiotic stewardship programs [16–18]. The multifaceted interventions needed to reduce infections and improve rational use of antibiotics have been difficult to sustain in high-income and low- and middle-income countries [15, 17, 19].

Given limited published studies on the sustainability of interventions for infection control and antibiotic stewardship, and the reasons why they were difficult to sustain the effect, we suggest 12 potential difficulties in sustaining the effect of a multifaceted intervention on infection control and antibiotic stewardship program, especially among those children in low- and middle-income countries based on our observations during this study. These include lack of performance feedback, staff turnover, message fatigue, distraction of other programs and initiatives, increased numbers of patients, increased complexity of higher-risk patients such as immune suppressed children, increased referred patients from outside hospitals, loosening of antibiotic prescribing standards over time, lack of consistency or intensity of input or reduction in frequency of training, lack of senior leadership input on this matter, never quite reaching the threshold of control of multi resistance organisms, and availability of resources such as hand hygiene and guidelines.

Previous studies indicate that quality improvement interventions need to be a continuous, flexible and evolving [16–19]. Staff turnover means that training and awareness of infection control and antibiotic stewardship program have to virtually be continuous. Continuous availability of guidelines, and standard antibiotics, and input from infection control teams as promoted by WHO guidelines in 2016 [20], need to be part of the health culture.

However, antibiotic prescribing is a complex process that is mutually and dependently influenced by patients, physicians, other healthcare providers, and the healthcare system. These most influential factors consist of intrinsic factors including physicians' attitudes (fear of undertreating a significant infection, complacency, or perverse incentives for poor prescribing) and extrinsic factors including patient-related factors (e.g. signs and symptoms) or healthcare system-related factors (e.g. time pressure and guidelines implemented) [21].

In some settings, a behavioral change approach on hand hygiene program using ongoing frequent audit and feedback with improvement of cognitive behavior and use of immediate

positive reinforcement has shown significant and sustained improvements in hand hygiene compliance at an acceptable cost [22].

## Recommendations for practice and research

Infection control programs should integrate two fundamental strategies to reduce healthcare-associated infections, consisting of reducing transmission of pathogens between patients and reducing the emergence and spread of antibiotic resistance.

We have shown that simple infection control measures including hand hygiene and rational use of antibiotics, are feasible and effective in the previous study in the same setting. To better understand why there was not a fully sustained effect, a qualitative study that explores the enablers and barriers to adhering to guidelines among the healthcare workers is needed. We considered that performance and compliance of the health workers on the infection control measures might need to be fed-back on a regular basis to ensure the sustainability of the program.

An antibiotic stewardship program should include prescribing behaviour in the recommendations, guidelines and policy, by pushing prescribers to make prescribing decisions that are beneficial both to the patient and to public health. Therefore, healthcare professionals need to be involved in the decision-making process to achieve and sustain the prescribing behaviour and optimal antibiotic prescribing [19].

Since the existing antibiotic stewardship program was not enough to achieve good long-term results, there is a need to develop a more comprehensive antibiotic stewardship program for healthcare workers both in PICU and pediatric wards. This includes ongoing education for healthcare workers on the rationale for and how to achieve rational use of antibiotic practice can reduce healthcare-associated infection related to antibiotic resistant bacteria.

## Strengths and Limitation

As far as we are aware, this study is the first study in a low-to-middle-income country to evaluate the sustainability of interventions to reduce healthcare-associated infections and irrational use of antibiotic in children, which is very common problem in such settings. A limitation of this study is a gap exists in the time period of intervention study in ending February 2013 and beginning February 2016 when the evaluation study was started, so that it is not continuous time period. We considered that ascertainment bias might occur in this study. But in this study, we addressed the potential for ascertainment bias in several ways. First, we did an identical data collection to diagnose healthcare-associated infections in both periods. Further, there were no changes in laboratory procedures between both periods that might falsify the culture results in the evaluation period tended to be positive. These relatively independent factors suggest that the identification of healthcare-associated infections between two periods was similar.

## Conclusions

This evaluation study showed the high burden of healthcare-associated infection and irrational use of antibiotics in children in our hospital remained even after implementing a multifaceted intervention to reduce healthcare-associated infection and irrational use of antibiotics. Possible reasons why it was difficult to sustain the effect of multifaceted intervention on infection control and antibiotic stewardship program to prevent healthcare-associated infections include lack of performance feedback or how and why were the antibiotic prescribing standards loosened. Ongoing surveillance and long-term monitoring of such programs at a level of intensity that includes weekly input is needed to sustain the effect in settings with limited

resources. The multifaceted infection control and antibiotic stewardship program should involve healthcare professionals in the decision-making processes, involve small group problem solving, and use positive reinforcement in order to sustain compliance and effectiveness.

## Supporting information

**S1 Data.**
(PDF)

**S2 Data.**
(XLSX)

## Acknowledgments

We would like to thank to Esta R Sativa and Infection Control and Antibiotic Resistance team at the Dr Sardjito Hospital, Yogyakarta, Indonesia.

## Author Contributions

**Conceptualization:** Indah K. Murni, Trevor Duke, Sharon Kinney, Andrew J. Daley, Ida S. Laksanawati, Nurnaningsih, Yati Soenarto.

**Data curation:** Indah K. Murni, M. Taufik Wirawan.

**Formal analysis:** Indah K. Murni, M. Taufik Wirawan.

**Investigation:** Indah K. Murni, Nurnaningsih, Desy Rusmawatiningtyas, M. Taufik Wirawan.

**Methodology:** Indah K. Murni, Trevor Duke, Sharon Kinney, Andrew J. Daley, M. Taufik Wirawan, Yati Soenarto.

**Project administration:** Desy Rusmawatiningtyas.

**Resources:** Indah K. Murni, Ida S. Laksanawati, Nurnaningsih.

**Supervision:** Trevor Duke, Sharon Kinney, Andrew J. Daley, Yati Soenarto.

**Validation:** Indah K. Murni, Trevor Duke, Ida S. Laksanawati.

**Writing – original draft:** Indah K. Murni.

**Writing – review & editing:** Indah K. Murni, Trevor Duke, Sharon Kinney, Andrew J. Daley, Ida S. Laksanawati, Nurnaningsih, Desy Rusmawatiningtyas, M. Taufik Wirawan, Yati Soenarto.

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
