## [Decision Letter · Decision Letter 0]

12 Nov 2019

PONE-D-19-24025

An evaluation of multifaceted interventions on hospital acquired infections and rational use of antibiotics in a developing country setting: Can they be sustained?

PLOS ONE

Dear Dr. Murni,

Thank you for submitting your manuscript to PLOS ONE. After careful consideration, we feel that it has merit but does not fully meet PLOS ONE’s publication criteria as it currently stands. Therefore, we invite you to submit a revised version of the manuscript that addresses the points raised during the review process.

The most important concern raised during the review process is the lack of description of interventions in place during the time periods that are compared. This information is critical to place the changes in HAI incidence in context. Please see specific comments and suggestions below. 

We would appreciate receiving your revised manuscript by Dec 27 2019 11:59PM. To enhance the reproducibility of your results, we recommend that if applicable you deposit your laboratory protocols in protocols.io, where a protocol can be assigned its own identifier (DOI) such that it can be cited independently in the future. For instructions see: http://journals.plos.org/plosone/s/submission-guidelines#loc-laboratory-protocols

We look forward to receiving your revised manuscript.

Kind regards,

Surbhi Leekha

Academic Editor

PLOS ONE

Journal Requirements:

Additional Editor Comments (if provided):

1. Please provide a table which describes the interventions and programs in place in the original post-intervention and in the follow-up evaluation phase. This should include (briefly) the composition/staffing of the infection control and antimicrobial stewardship programs in each phase. This information is critical to help the reviewer understand the potential reasons for increases in HAIs

2. The basis for Table 5 is unclear: is this based on the literature or your observations in your facility and program? Recommend deleting, and including some of these elements in the Table describing the interventions and program in each phase (e.g., senior leadership input and support, resources for hand hygiene etc.)

3. The type of regression analyses used needs to be specified.

4. Tables should have more descriptive and meaningful titles. For example, Table 1. Baseline characteristics of ?? in post-intervention and evaluation period. Add footnotes defining post-intervention and evaluation period with dates added to the column headers or footnote. All tables with measures of relative risk/odds ratio should have the statistical method used to derive those also stated as table footnote.

Reviewers' comments:

Reviewer's Responses to Questions

**Comments to the Author**

1. Is the manuscript technically sound, and do the data support the conclusions?

Reviewer #1: Yes

Reviewer #2: Partly

2. Has the statistical analysis been performed appropriately and rigorously? 

Reviewer #1: I Don't Know

Reviewer #2: Yes

3. Have the authors made all data underlying the findings in their manuscript fully available?

Reviewer #1: Yes

Reviewer #2: Yes

4. Is the manuscript presented in an intelligible fashion and written in standard English?

Reviewer #1: Yes

Reviewer #2: No

5. Review Comments to the Author

Reviewer #1: Overall, this is an important topic addressed by authors-- maintenance of HAI and antimicrobial stewardship interventions after an initial intervention. Clearly, the authors have put a lot of work into getting this data and reporting upon it in this hospital in Indonesia.

I have several comments that I think would overall strengthen the paper and make it more likely to meet requirements to be published in something like PLOS One.

Abstract: It is unclear in the abstract what the actual intervention was and what was done in the maintenance phase -- this is unclear in the paper as well. This should be delineated in the abstract for the reader.

Throughout the paper, I would use healthcare-associated infection rather than hospital acquired and rather than nosocomial as per current HAI lingo.

Methods: the methods should clearly outline the interventions and the surveillance interventions --after reading and re reading the paper, I'm not sure what happened during the intervention versus what happened in the maintenance phase. More helpful than the tables presented would be a timeline of the initial intervention and what continued in the maintenance phase. (would replace table 5 which should all be in the discussion instead with this timeline instead)

In the methods, the authors defined inappropriate duration of antibiotics as more than 20% longer than the recommended duration (according to what? and also have other studies used this definition? if yes, would cite.)

In methods, I'd like to know more details about the IC and Abx stewardship team. Who made up each team? Was there a MD on both? What members are on the stewardship team, if there isnt a team then on the committee. Did anyone have protected time for IC or for stewardship or get money for these roles?

These details are important in international IC and stewardship studies.

Results:

Would give overall HAI but also individual HAI numbers as possible (which HAIs were you looking at, which seemed to increase the most?-- would add a lot to the paper)

For hand hygiene compliance, would add the total number of observations into the paper.

Is there a way to measure case mix index in the two periods for the mortality results?

Discussion

The discussion is wordy but doesn't get at the meat of the paper.

Again, I was struck that I didnt understand the intervention vs maintenance phase well--- this needs to be much clearer in this paper and will help frame the discussion.

What pieces of the maintenance phase if any do authors think are helping, what should be strengthened, what could particularly help if the leadership was going to invest into one specific area?

How could the CDC core elements of stewardship for limited resource settings help? This whole discussion should be streamlined and identify things that could happen in the maintenance phase that would be high yield (would do some of a lit search to help inform this)

Reviewer #2: The manuscript entitled “an evaluation of multifaceted interventions on HAI and rational use of antibiotics in a developing country setting: can they be sustained?” is a prospective cohort observational study and a follow-up of a study published in 2015.

The authors conducted surveillance of HAI and AMU 5 years after implementing an infection control bundle and a stewardship program at hospital in Indonesia.

The initial 2015 study had shown benefits of this bundle in terms of outcomes: the authors noted decreased incidence of HAI, decreased mortality, and a higher proportion of appropriate AMU (though no decrease in total AMU). In this follow-up study, they note that rates of HAI and inappropriate AMU had increased (close to the baseline pre-intervention rates), though mortality was stable or had slightly decreased.

In this reviewer's opinion the topic of sustainability (and cost-effectiveness) of interventions to reduce HAI and AMR is very pertinent for LMICs, and certainly worthy of publication. The authors should be commended for tackling this follow-up study, after significants efforts to implement the intervention some. years ago. Though this paper has merit, because it mostly describes follow-up surveillance data (using identical methods and analysis and without additional interventions), in this reviewer's opinion a concise communication might be preferable to a full text article.

Additional concerns are provided in more detail below, in the hope they can be helpful for the authors.

Major concerns:

- In the 2015 paper from the same authors, the intervention appears to be an “intensive intervention with HH, educational sessions, and audit-feedback for antimicrobial use (AMU)” for 3 months, then a period of “post-intervention” during which effectiveness of the bundle was measured in terms of incidence of HAI and AMU. In this paper, the authors have compared the data from the previous study (post-intervention phase)with data from the follow-up phase, but what is missing is data on the sort of activities which were (or were not) occurring. The paper is entitled “sustainability of interventions” , but the data provided that does not clearly support that it was the absence of interventions that led to the increased rates. Few details are provided on monitoring of activities: what sort of indicators were used to determine that certain activities were sustained but not others (eg: KPI?). It would also be helpful to clarify whether the research team was collecting surveillance data prospectively or if surveillance had become fully incorporated into routine quality activities, and was conducted by a different group of individuals in the follow-up phase. The discussion alludes to the fact that the hospital culture had changed and some activities were conducted, but audit-feedback activities were no longer conducted. Was this also not the case in the initial post-intervention phase (2015?). This whole aspect is somewhat confusing and should be clarified in the methods section – or, the manuscript significantly shortened to a concise communication simply stating that 5 years after implementing a bundle consisting of Infection Control Program and a Stewardship program, rates of HAI had increased.

- HAI rates increased but mortality decreased in the follow-up phase: the authors argue this may be due to overall improved quality of care, but they also attribute increased HAI rates to worse practices and high staff turnover, so there is an inherent contradiction. Further, no details are provided on whether this was all-cause mortality, in hospital mortality, 30-day mortality, etc and most importantly, what was the mortality attributable to infection in both these groups? is it possible that mortality was lower because of the use of broad spectrum antibiotics? Was selection/ascertainment bias involved in this?

- Since a lot of effort went into identifying HAI based on microbiology and clinical findings, data on characteristics and types of HAI (number of cases of HAP, UTI, CLABSI etc) (as well as a description of who did the surveillance) would be helpful to better understand shifts in epidemiology and increasing rates over time.

- Also, it appears antibiograms were collected every 6 months: if available, this data should be provided (at least for some organisms that are relevant for HAI) so we can assess evolving rates of drug-resistance over time, and attempt to correlate with AMU, mortality etc.

- The discussion is largely speculative, and lists solutions that are not linked to the findings of the study. it would be more useful to point to specific lessons learned that other centres in LMICs can learn from, rather than provide a list of recommendations and best practices which are very generic. For example, did the group identify specific barriers to sustaining audit-feedback over time? The authors should elaborate much more on the limitations of the study – in particular discuss possible selection and ascertainment bias.

- Table 5 is unnecessary and repeats much of what was said in discussion.

- The reference that was provided for treatment guidelines is the 2005 WHO guideline: was that really the guideline used by clinicians in the hospital until 2018? If so, the impact of outdated guidelines on clinical practice should be addressed and the authors should consider revising their assessment of appropriate vs inappropriate antibiotic prescriptions?

- Same comment for the CLSI guidelines (2007) – is that really what the laboratory was using? These are updated every year with some major revisions in susceptibility breakpoints in the past 12 years. Please review.

Minor points

- An operational definition for the 2 periods should be provided and specify exactly the duration of each, as well as what activities were occurring when

- Table 2, 3 and 4 should not be entitled “sustainability of intervention..” which is an interpretation – the titles should be neutral and refer to what is shown (eg: rates of HAI over the study period, ..

The term "developing country" should be revised to the more appropriate LMIC, or other term that is now more in use, and used consistently throughout the text.

- The text should be revised for clarity and flow, and some sentences should be re-structured.

6. PLOS authors have the option to publish the peer review history of their article (what does this mean?). If published, this will include your full peer review and any attached files.

Reviewer #1: No

Reviewer #2: No

---

## [Author Response · Author response to Decision Letter 0]

31 Dec 2019

Response to Reviewer

 Journal Requirements:  1. When submitting your revision, we need you to address these additional requirements.

Response: Thank you so much. Yes, we have revised accordingly.

Response: Yes, we have added the information about patient consent because we only collected routine data without intervening. The ethics committee did not require individual patient consent, but all parents of children were informed of the study.

 

Additional Editor Comments (if provided):  1. Please provide a table which describes the interventions and programs in place in the original post-intervention and in the follow-up evaluation phase. This should include (briefly) the composition/staffing of the infection control and antimicrobial stewardship programs in each phase. This information is critical to help the reviewer understand the potential reasons for increases in HAIs

Response: Yes, we have added a table which describes the interventions in each period of the study.

2. The basis for Table 5 is unclear: is this based on the literature or your observations in your facility and program? Recommend deleting, and including some of these elements in the Table describing the interventions and program in each phase (e.g., senior leadership input and support, resources for hand hygiene etc.)

Response: The table was based on our observations in my facility and program, we considered to delete and included these elements in the Table describing intervention and in the Discussion section. 

3. The type of regression analyses used needs to be specified.

Response: We did not use regression analysis in this analysis and we have deleted this from the manuscript. 

4. Tables should have more descriptive and meaningful titles. For example, Table 1. Baseline characteristics of ?? in post-intervention and evaluation period. Add footnotes defining post-intervention and evaluation period with dates added to the column headers or footnote. All tables with measures of relative risk/odds ratio should have the statistical method used to derive those also stated as table footnote. 

Response: Yes, we have revised the description of Table 1. We have added footnotes about the relative risk (RR) that was calculated to compare the effect of the interventions between both periods by calculating the ratio of the probability of an outcome in the post-intervention period to the probability of an outcome in the evaluation period, the time of both periods.   

 5. Review Comments to the Author  

Please use the space provided to explain your answers to the questions above. You may also include additional comments for the author, including concerns about dual publication, research ethics, or publication ethics. (Please upload your review as an attachment if it exceeds 20,000 characters)  

Reviewer #1: Overall, this is an important topic addressed by authors-- maintenance of HAI and antimicrobial stewardship interventions after an initial intervention. Clearly, the authors have put a lot of work into getting this data and reporting upon it in this hospital in Indonesia. I have several comments that I think would overall strengthen the paper and make it more likely to meet requirements to be published in something like PLOS One.  Abstract: It is unclear in the abstract what the actual intervention was and what was done in the maintenance phase -- this is unclear in the paper as well. This should be delineated in the abstract for the reader.

Response: Thank you so much for the suggestions. 

We have provided a table which describes the interventions in each period (Table 1).  

Throughout the paper, I would use healthcare-associated infection rather than hospital acquired and rather than nosocomial as per current HAI lingo.

Y

Response: Yes, we have revised accordingly.  

Methods: the methods should clearly outline the interventions and the surveillance interventions --after reading and re reading the paper, I'm not sure what happened during the intervention versus what happened in the maintenance phase. More helpful than the tables presented would be a timeline of the initial intervention and what continued in the maintenance phase. (would replace table 5 which should all be in the discussion instead with this timeline instead)

Response: Yes, we have provided a table which describes the interventions during the intervention, the post-intervention, and the evaluation periods. The content of this table was based on our observations in my facility and program, we considered to delete Table 5. We decided to include these to the Discussion section in order to explain the reason why it was difficult to sustain the effect of interventions. 

In the methods, the authors defined inappropriate duration of antibiotics as more than 20% longer than the recommended duration (according to what? and also have other studies used this definition? if yes, would cite.)

Response: We used this criteria in our previous study (the intervention study). 

In methods, I'd like to know more details about the IC and Abx stewardship team. Who made up each team? Was there a MD on both? What members are on the stewardship team, if there isnt a team then on the committee. Did anyone have protected time for IC or for stewardship or get money for these roles? These details are important in international IC and stewardship studies.

The infection control and antibiotic stewardship team consist of infectious disease doctors, specialist doctors, nurses, the pharmacist and clinical pathology and microbiology staff. We have no protected time for infection control and antibiotic stewardship nor get money for these roles.   Results: Would give overall HAI but also individual HAI numbers as possible (which HAIs were you looking at, which seemed to increase the most?-- would add a lot to the paper)

Response: We considered to publish this separately. For hand hygiene compliance, would add the total number of observations into the paper. The total number of observation in the evaluation period was 2993 (Table 5). Overall hand hygiene compliance among the healthcare workers in the evaluation period decreased from 62.9% (1125/1789) to 51% (1526/2993) (p<0.001)

Is there a way to measure case mix index in the two periods for the mortality results?

Response: In this present study, we did not apply the case mix risk adjusted in-hospital mortality.

Discussion

 The discussion is wordy but doesn't get at the meat of the paper. Again, I was struck that I didnt understand the intervention vs maintenance phase well--- this needs to be much clearer in this paper and will help frame the discussion. What pieces of the maintenance phase if any do authors think are helping, what should be strengthened, what could particularly help if the leadership was going to invest into one specific area?

Response: We have provided Table 1 of interventions in each period of the study. We considered that performance and compliance of the health workers on the infection control measures might need to be feed back to them in a regular basis to ensure the sustainability of the program.

 

How could the CDC core elements of stewardship for limited resource settings help? This whole discussion should be streamlined and identify things that could happen in the maintenance phase that would be high yield (would do some of a lit search to help inform this)

Response: We did a systematic review on the prevention of HAI which concludes that hand hygiene and antibiotic stewardship program could reduce HAI up to 50%. Therefore, these measures should be conducted simultaneously. But performance and compliance of the health workers on these measures might need to be feed back to them in a regular basis to ensure the sustainability of the program (Paediatr Int Child Health 2013;33(2):61-78).

  

Reviewer #2: The manuscript entitled “an evaluation of multifaceted interventions on HAI and rational use of antibiotics in a developing country setting: can they be sustained?” is a prospective cohort observational study and a follow-up of a study published in 2015. The authors conducted surveillance of HAI and AMU 5 years after implementing an infection control bundle and a stewardship program at hospital in Indonesia. The initial 2015 study had shown benefits of this bundle in terms of outcomes: the authors noted decreased incidence of HAI, decreased mortality, and a higher proportion of appropriate AMU (though no decrease in total AMU). In this follow-up study, they note that rates of HAI and inappropriate AMU had increased (close to the baseline pre-intervention rates), though mortality was stable or had slightly decreased.  In this reviewer's opinion the topic of sustainability (and cost-effectiveness) of interventions to reduce HAI and AMR is very pertinent for LMICs, and certainly worthy of publication. The authors should be commended for tackling this follow-up study, after significants efforts to implement the intervention some. years ago. Though this paper has merit, because it mostly describes follow-up surveillance data (using identical methods and analysis and without additional interventions), in this reviewer's opinion a concise communication might be preferable to a full text article.  Additional concerns are provided in more detail below, in the hope they can be helpful for the authors.  Major concerns:  - In the 2015 paper from the same authors, the intervention appears to be an “intensive intervention with HH, educational sessions, and audit-feedback for antimicrobial use (AMU)” for 3 months, then a period of “post-intervention” during which effectiveness of the bundle was measured in terms of incidence of HAI and AMU. In this paper, the authors have compared the data from the previous study (post-intervention phase) with data from the follow-up phase, but what is missing is data on the sort of activities which were (or were not) occurring. The paper is entitled “sustainability of interventions”, but the data provided that does not clearly support that it was the absence of interventions that led to the increased rates. Few details are provided on monitoring of activities: what sort of indicators were used to determine that certain activities were sustained but not others (eg: KPI?). It would also be helpful to clarify whether the research team was collecting surveillance data prospectively or if surveillance had become fully incorporated into routine quality activities, and was conducted by a different group of individuals in the follow-up phase. The discussion alludes to the fact that the hospital culture had changed and some activities were conducted, but audit-feedback activities were no longer conducted. Was this also not the case in the initial post-intervention phase (2015?). This whole aspect is somewhat confusing and should be clarified in the methods section – or, the manuscript significantly shortened to a concise communication simply stating that 5 years after implementing a bundle consisting of Infection Control Program and a Stewardship program, rates of HAI had increased.

Response: Thank you so much for the suggestion. Yes, the research team was collecting surveillance data prospectively, but the surveillance had not become fully incorporated into routine quality activities in the post-intervention and follow-up phase. The hospital did not have team to do active surveillance, they collect data by doing passive surveillance of healthcare-associated infection but not routinely. The hospital did not do surveillance of antibiotic use and rational use of antibiotics. The hospital culture had changed and some activities were conducted, but audit-feedback activities were no longer conducted. This was also the case in the initial post-intervention phase.

- HAI rates increased but mortality decreased in the follow-up phase: the authors argue this may be due to overall improved quality of care, but they also attribute increased HAI rates to worse practices and high staff turnover, so there is an inherent contradiction. Further, no details are provided on whether this was all-cause mortality, in hospital mortality, 30-day mortality, etc and most importantly, what was the mortality attributable to infection in both these groups? is it possible that mortality was lower because of the use of broad spectrum antibiotics? Was selection/ascertainment bias involved in this?

Response: The mortality rate was all-cause in hospital mortality. Mortality in children with healthcare-associated infection was 4.1% (13/314). Mortality related to healthcare-associated infection was markedly lower in this evaluation period compared to the mortality in the post-intervention period of 24.5%. 

We considered that ascertainment bias might occur in this study. But in this study, we addressed the potential for ascertainment bias in several ways. First, we did an identical data collection to diagnose healthcare-associated infections in both periods. Further, there were no changes in laboratory procedures between both periods that might falsify the culture results in the evaluation period tended to be positive. These relatively independent factors suggest that the identification of healthcare-associated infections between two periods was similar.

 - Since a lot of effort went into identifying HAI based on microbiology and clinical findings, data on characteristics and types of HAI (number of cases of HAP, UTI, CLABSI etc) (as well as a description of who did the surveillance) would be helpful to better understand shifts in epidemiology and increasing rates over time.

Response: The surveillance of HAI was done by the research team not incorporated with the hospital program. The detailed HAI in the evaluation period will be published later, but the incidence of nosocomial bloodstream infection in the post intervention period was 92/1419 (6.5%) among all recruited children and 92/123 (74.8%) among children with HAI, while in the evaluation period was 36/1885 (1.9%) among all recruited children or 36/314 (11.5%) among children with HAI. We considered that although the proportion of HAI increased, the severity of HAI decreased. So that the mortality associated with HAI also decreased, from 25.5% to 4.1%, in the post-intervention period and the evaluation period, respectively. 

- Also, it appears antibiograms were collected every 6 months: if available, this data should be provided (at least for some organisms that are relevant for HAI) so we can assess evolving rates of drug-resistance over time, and attempt to correlate with AMU, mortality etc.

Response: In the post-intervention period, the most common organism related to HAI was Pseudomonas aeruginosa, while in the evaluation period was Klebsiella pneumoniae.

- The discussion is largely speculative, and lists solutions that are not linked to the findings of the study. it would be more useful to point to specific lessons learned that other centres in LMICs can learn from, rather than provide a list of recommendations and best practices which are very generic. For example, did the group identify specific barriers to sustaining audit-feedback over time? The authors should elaborate much more on the limitations of the study – in particular discuss possible selection and ascertainment bias.

Response: We considered that ascertainment bias might occur in this study. But in this study, we addressed that issue of ascertainment bias in several ways. First, we did an identical data collection to diagnose healthcare-associated infections in both periods. Further, there were no changes in laboratory procedures between both periods that might falsify the culture results in the evaluation period tended to be positive. These relatively independent factors suggest that the identification of healthcare-associated infections between two periods was similar. 

- Table 5 is unnecessary and repeats much of what was said in discussion.

Response: Yes, we have deleted Table 5 and included in the discussion.

- The reference that was provided for treatment guidelines is the 2005 WHO guideline: was that really the guideline used by clinicians in the hospital until 2018? If so, the impact of outdated guidelines on clinical practice should be addressed and the authors should consider revising their assessment of appropriate vs inappropriate antibiotic prescriptions?

Response: We have updated the guidelines which used the 2013 WHO guideline. 

 - Same comment for the CLSI guidelines (2007) – is that really what the laboratory was using? These are updated every year with some major revisions in susceptibility breakpoints in the past 12 years. Please review.

Response: We have updated the CLSI guidelines (2015). 

CLSI. Performance standards for antimicrobial susceptibility testing-seventeeth informationan supplement. CLSI document M07-A10. 2015.

 Minor points - An operational definition for the 2 periods should be provided and specify exactly the duration of each, as well as what activities were occurring when

The post-intervention era was from 1 December 2011 to 28 February 2013 and the evaluation evaluation era was started in 1 February 2016 to 30 April 2018. The activities were occurring in each period was provided in Table 1. 

- Table 2, 3 and 4 should not be entitled “sustainability of intervention..” which is an interpretation – the titles should be neutral and refer to what is shown (eg: rates of HAI over the study period, .. The term "developing country" should be revised to the more appropriate LMIC, or other term that is now more in use, and used consistently throughout the text.

Response: Yes, we have revised accordingly.

- The text should be revised for clarity and flow, and some sentences should be re-structured.

Response: Yes, we have revised accordingly.

---

## [Editor Report · Decision Letter 1]

13 Jan 2020

PONE-D-19-24025R1

Multifaceted interventions for healthcare-associated infections and rational use of antibiotics in a low-to-middle-income country: Can they be sustained?

PLOS ONE

Dear Dr. Murni,

Thank you for submitting your manuscript to PLOS ONE. After careful consideration, we feel that it has merit but does not fully meet PLOS ONE’s publication criteria as it currently stands. Therefore, we invite you to submit a revised version of the manuscript that addresses the points raised during the review process.

Thank you for making the suggested changes to your manuscript. Please address the following additional comments:

1. Under Methods section A. Hospital-acquired infection, the authors state that “The definitions of HAI were based on the US Centers for Disease Control and Prevention (CDC) National Healthcare Safety Network (NHSN). Every child in the study was observed each day to see whether he/she fulfilled the CDC criteria for a HAI. If criteria were fulfilled, the treating medical and nursing staff were advised to collect a culture of blood, urine or other sterile sites as appropriate on the same day.”

Please elaborate on this process – how many/what proportion of culture collection was guided by the study surveillance vs. based on clinical suspicion of infection? Could the variability in this process explain any of the observed increase in HAIs during the maintenance phase?

2. The authors have clarified that regression analysis was not used and there was no accounting for changes in case-mix over time. I would include in the discussion as a limitation of an analysis that it does not account for potential changes in patient characteristics over time. Please also consider a brief statement on whether any major changes in patient characteristics occurred to your knowledge e.g., change in immunocompromised host population, types of surgery etc.

3. Please proof-read to ensure that HAI is now spelled out as healthcare-associated infection throughout (some instances of use of hospital-acquired were noted)

4. Change header stating “The use of irrational antibiotics” to “The irrational use of antibiotics”

We would appreciate receiving your revised manuscript by Feb 27 2020 11:59PM. To enhance the reproducibility of your results, we recommend that if applicable you deposit your laboratory protocols in protocols.io, where a protocol can be assigned its own identifier (DOI) such that it can be cited independently in the future. For instructions see: http://journals.plos.org/plosone/s/submission-guidelines#loc-laboratory-protocols

We look forward to receiving your revised manuscript.

Kind regards,

Surbhi Leekha

Academic Editor

PLOS ONE

---

## [Author Response · Author response to Decision Letter 1]

27 Feb 2020

Dear Reviewers and Editor,

We have revised the manuscript accordingly as suggested. We have addressed the following additional comments:

1. Under Methods section A. Hospital-acquired infection, the authors state that “The definitions of HAI were based on the US Centers for Disease Control and Prevention (CDC) National Healthcare Safety Network (NHSN). Every child in the study was observed each day to see whether he/she fulfilled the CDC criteria for a HAI. If criteria were fulfilled, the treating medical and nursing staff were advised to collect a culture of blood, urine or other sterile sites as appropriate on the same day.”

Please elaborate on this process – how many/what proportion of culture collection was guided by the study surveillance vs. based on clinical suspicion of infection? Could the variability in this process explain any of the observed increase in HAIs during the maintenance phase?

Response to the Reviewer:

We did not have data detailing which cultures were collected based on the study surveillance definition or based on clinical suspicion of infection of the treating doctors. We included those two as one. 

But we did an identical data collection to diagnose nosocomial infections in both periods. Those cultures were taken when patients had signs and symptoms of infection. Further, there were no changes in laboratory procedures between the post intervention and maintenance periods that might falsify the culture results. These relatively independent factors suggest that the identification of nosocomial infections between two periods was similar. 

2. The authors have clarified that regression analysis was not used and there was no accounting for changes in case-mix over time. I would include in the discussion as a limitation of an analysis that it does not account for potential changes in patient characteristics over time. Please also consider a brief statement on whether any major changes in patient characteristics occurred to your knowledge e.g., change in immunocompromised host population, types of surgery etc.

Response to the Reviewer:

We decided to perform multivariable logistic regression analysis to quantify the relationship between the HAI and the multifaceted intervention allowing for statistical control of potential confounders including age, sex, the presence of syndrome, immunocompromised condition, referral patient, sepsis, and malnutrition. We add the Table 4. Multivariable analysis of the factors affecting intervention on HAI.

3. Please proof-read to ensure that HAI is now spelled out as healthcare-associated infection throughout (some instances of use of hospital-acquired were noted)

Response to the Reviewer: We have revised accordingly.

4. Change header stating “The use of irrational antibiotics” to “The irrational use of antibiotics”

Response to the Reviewer: We have revised accordingly.

---

## [Editor Report · Decision Letter 2]

27 Apr 2020

PONE-D-19-24025R2

Multifaceted interventions for healthcare-associated infections and rational use of antibiotics in a low-to-middle-income country: Can they be sustained?

PLOS ONE

Dear Dr. Murni,

Thank you for submitting your manuscript to PLOS ONE. After careful consideration, we feel that it has merit but does not fully meet PLOS ONE’s publication criteria as it currently stands. Therefore, we invite you to submit a revised version of the manuscript that addresses the points raised during the review process.

1. Thank you for clarifying the process of culture collection. Please include this information and explanation in the methods section. 

2. Thank you for adding the multivariable regression analysis. Please also do the following: a) ensure that variable names in the model are the same as the variable names in Table 2. b) Table 4 is not clear, and  likely not needed. It would be better to describe the model building process in the text itself - were the variables added stepwise, what were the criteria for including the variables in the model etc. c) The statement "We did a multivariable analysis to adjust for differences in patient characteristics." is not necessary in the Results section, it is already explained in the Methods. 

3) In the discussion, when discussing major findings, I would state that there were no major differences in the patient population in the two time periods, and even after adjusting for differences, there was an increase in HAIs. Similarly, I would expand on the statement "Staff turnover and health care shortages create problems

because new doctors and nurses were not aware of infection control bundle especially in PICU." Do you think this was the major reason for the increase, please state this more clearly and expand with a couple of additional statements, 

We would appreciate receiving your revised manuscript by Jun 11 2020 11:59PM. To enhance the reproducibility of your results, we recommend that if applicable you deposit your laboratory protocols in protocols.io, where a protocol can be assigned its own identifier (DOI) such that it can be cited independently in the future. For instructions see: http://journals.plos.org/plosone/s/submission-guidelines#loc-laboratory-protocols

We look forward to receiving your revised manuscript.

Kind regards,

Surbhi Leekha

Academic Editor

PLOS ONE

---

## [Author Response · Author response to Decision Letter 2]

7 May 2020

Dear Reviewers and Editor,

We have revised the manuscript accordingly as suggested. We have addressed the following additional comments:

1. Thank you for clarifying the process of culture collection. Please include this information and explanation in the methods section. 

Response to the Reviewer:

Thank you. Yes, we have included this information in the Methods section. 

We did not have data detailing which cultures were collected based on the study surveillance definition or based on clinical suspicion of infection of the treating doctors. We included those two as one. But we did an identical data collection to diagnose nosocomial infections in both periods. Those cultures were taken when patients had signs and symptoms of infection.

2. Thank you for adding the multivariable regression analysis. Please also do the following: a) ensure that variable names in the model are the same as the variable names in Table 2. b) Table 4 is not clear, and likely not needed. It would be better to describe the model building process in the text itself - were the variables added stepwise, what were the criteria for including the variables in the model etc. c) The statement "We did a multivariable analysis to adjust for differences in patient characteristics." is not necessary in the Results section, it is already explained in the Methods. 

Response to the Reviewer:

a. Thank you, we have revised the variable name in the Table 2 is the same as the variable in the model.

b. We have deleted Table 4 and described the model in the text.

c. Yes, we have deleted and revised accordingly.

3) In the discussion, when discussing major findings, I would state that there were no major differences in the patient population in the two time periods, and even after adjusting for differences, there was an increase in HAIs. Similarly, I would expand on the statement "Staff turnover and health care shortages create problems

because new doctors and nurses were not aware of infection control bundle especially in PICU." Do you think this was the major reason for the increase, please state this more clearly and expand with a couple of additional statements

Response to the Reviewer:

Thank you, we have added the statement as suggested. 

Staff turnover and health care shortages create problems because new doctors and nurses were not aware of infection control bundle especially in PICU. Increasing nurse-to-patient ratios have been associated with the spread of infection. Further, nurse shortages are also related to increased risk of healthcare-associated infections.

---

## [Editor Report · Decision Letter 3]

22 May 2020

Multifaceted interventions for healthcare-associated infections and rational use of antibiotics in a low-to-middle-income country: Can they be sustained?

PONE-D-19-24025R3

Dear Dr. Murni,

We are pleased to inform you that your manuscript has been judged scientifically suitable for publication and will be formally accepted for publication once it complies with all outstanding technical requirements.

With kind regards,

Surbhi Leekha

Academic Editor

PLOS ONE
---

## [Editor Report · Acceptance letter]

4 Jun 2020

PONE-D-19-24025R3 

Multifaceted interventions for healthcare-associated infections and rational use of antibiotics in a low-to-middle-income country: Can they be sustained? 

Dear Dr. Murni:

I'm pleased to inform you that your manuscript has been deemed suitable for publication in PLOS ONE. Congratulations! Your manuscript is now with our production department. 

Kind regards, 

on behalf of

Dr. Surbhi Leekha 

Academic Editor

PLOS ONE